# Attribute Prediction as Multiple Instance Learning

**Diego Marcos** *diego.marcos@inria.fr*
*Inria, France*

**Aike Potze** *aike.potze@wur.nl*
*Wageningen University, The Netherlands*

**Wenjia Xu** *xuwenjia16@mails.ucas.ac.cn*
*Chinese Academy of Sciences, China*

**Devis Tuia** *devis.tuia@epfl.ch*
*EPFL, Switzerland*

**Zeynep Akata** *zeynep.akata@uni-tuebingen.de*
*University of Tübingen, Germany*

**Reviewed on OpenReview:** *https://openreview.net/forum?id=nmFczdJtc2*

## Abstract

Attribute-based representations help machine learning models perform tasks based on human understandable concepts, allowing a closer human-machine collaboration. However, learning attributes that accurately reflect the content of an image is not always straightforward, as per-image ground truth attributes are often not available. We propose applying the Multiple Instance Learning (MIL) paradigm to attribute learning (AMIL) while only using class-level labels. We allow the model to under-predict the positive attributes, which may be missing in a particular image due to occlusions or unfavorable pose, but not to over-predict the negative ones, which are almost certainly not present. We evaluate it in the zero-shot learning (ZSL) setting, where training and test classes are disjoint, and show that this also allows to profit from knowledge about the semantic relatedness of attributes. In addition, we apply the MIL assumption to ZSL classification and propose MIL-DAP, an attribute-based zero-shot classification method, based on Direct Attribute Prediction (DAP), to evaluate attribute prediction methods when no image-level data is available for evaluation. Experiments on CUB-200-2011, SUN Attributes and AwA2 show improvements on attribute detection, attribute-based zero-shot classification and weakly supervised part localization.

## 1 Introduction

Semantic attributes provide a rich, human understandable representation that allows to connect the visual world with other sources of knowledge, such as text corpora and knowledge graphs (Rohrbach et al., 2010). This enables automatic reasoning that uses a set of pre-defined semantic elements (Hudson & Manning, 2019), allowing a much closer collaboration between these systems and humans (Branson et al., 2010). Attribute-based representations have proven useful for domain adaptation (Han et al., 2014; Chen et al., 2015; Gebru et al., 2017), natural scene perception (Marcos et al., 2020), image-to-image translation (Wu et al., 2019), image retrieval (Siddiquie et al., 2011; Felix et al., 2012), and zero-shot learning (ZSL) (Lampert et al., 2013). Attribute annotations are often obtained by asking human annotators to either provide image-level attributes (Patterson & Hays, 2012; Wah et al., 2011), which is labour intensive and requires all attributes, or at least a few (Sun et al., 2015), to be annotated in every image, or class-level attribute labels (Lampert et al., 2013), requiring as many annotations as there are classes. An alternative approach that requires

no manual annotations is to use class ontologies such as WordNet (Al-Halah & Stiefelhagen, 2015) or text corpora (Al-Halah & Stiefelhagen, 2017) in order to mine class-level attributes.

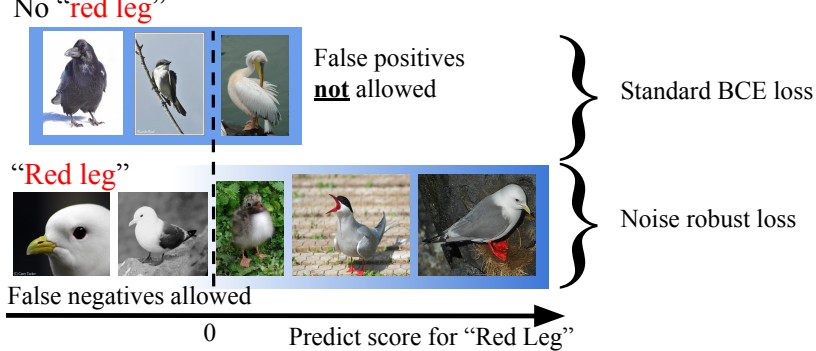

Figure 1: We formulate the task of attribute prediction as a dataset-wide MIL framework, where some classes *may* contain a given attribute (positive bag) and some *must* not contain it (negative bag). We suggest treating the positives differently during training by allowing false negatives using a noise robust loss.

These class-level attributes inform us about the attributes that are potentially present in an image of a given class, but do not help us in knowing whether the attribute is actually visible on a specific image. For instance, if we know that an image contains an Arctic tern or a Red-legged kittiwake (seen in Figure 1), we can conclude that it is possible, although not certain, that the image contains red legs, since they may be occluded, for instance. On the other hand, knowing the class also provides us with information about attributes that will surely *not* be present: if the image is labeled as Red-legged kittiwake, we can almost certainly conclude that there will be no yellow legs visible. We propose to use this asymmetry between positive and negative class-level attributes to improve image-level attribute prediction.

Our solution is based on the Multiple Instance Learning (MIL) (Dietterich et al., 1997; Maron & Lozano-Pérez, 1998) paradigm, i.e. weakly supervised learning in which samples are distributed in *bags* and labels are provided only at the bag level. A bag labeled as negative means that *all* the samples it contains are negative; a bag labeled as positive means that *at least one* sample in the bag is positive. We build our bags using: 1) images with single class labels, e.g. animal species, and 2) class-level attribute annotations in the form of a binary matrix representing which attribute is possibly present in each class. Using these two elements, we cast attribute learning as a multi-label MIL problem, or MIML (Zhou et al., 2012). In contrast with previous approaches, in which the size of the bags is limited by their memory footprint, ours can handle bags of unlimited size by applying a noise robust loss to the positive bags and a standard binary cross entropy loss to the negatives (Figure 1). We show that this approach is also particularly suitable for injecting additional external knowledge in the form of attribute relatedness.

Under the assumption that only class-level attribute annotations are available, we run into the problem of evaluating the quality of the attribute predictions without resorting to image-level attributes. To solve this issue, we propose MIL-DAP, an attribute-based ZSL method that allows to evaluate attribute prediction quality without image-level labels. We test the proposed Attribute MIL (AMIL) approach on three attribute datasets (CUB-200-2011, SUN Attributes and AwA2) and consistently show that we are able to improve the performance on attribute detection, attribute-based zero-shot classification and weakly supervised part localization.

## 2    Related works

**MIL in computer vision.**   MIL approaches often build each bag using all the patches or region proposals stemming from an individual image: the label is provided at image-level but the desired solution is at patch-level. This has been applied to weakly supervised object detection (Wang & Forsyth, 2009; Wan et al., 2019), semantic segmentation (Pathak et al., 2015), attention-based classification (Ilse et al., 2018) and Zero-Shot Learning (ZSL) (Rahman & Khan, 2018). When accounting for multiple attributes, MIL not only results

in multiple overlapping bags, a setting called Multiple Instance Multi-Label setting, MIML (Zhou et al., 2012), but, in addition, each bag spans almost the entire dataset, since for each attribute the positive and negative bags jointly contain all classes and thus the whole dataset. This setting is not appropriate for traditional MIL approaches, since these tend to require performing computations over all instances within a bag, thus practically limiting the bag size. A possible solution is to approximate the MIL function by building bags at the batch level (Huang et al., 2014; Feng & Zhou, 2017), alleviating the memory issue, or to use dynamic programming to make the problem more tractable (Pham et al., 2017). Our proposed AMIL implicitly keeps track of dataset-wide statistics by applying a noise robust loss to the positive attributes that penalizes false negatives less strictly than false positives, thus deciding on whether or not the positive labels should be enforced by comparing them to the current decision boundary while allowing the use of batch-based methods.

**Learning with noisy labels.** Some approaches use auxiliary information, such as a small set of clean labels, to model the relation between noisy and clean labels for class correction (Vahdat, 2017) or to learn a re-weighting of the noisy samples minimizing a loss on the clean samples (Ren et al., 2018; Zhang et al., 2020). Other works use the current predictions of the model to find wrongly labeled samples. They aim at reducing the high loss imposed to instances confidently misclassified by the model, either by weighing the loss with the confidence of the model (Mnih & Hinton, 2012), linearly combining the label with the current prediction (Reed et al., 2015) or by making the penalization of large errors tunable to adapt to different noise levels (Zhang & Sabuncu, 2018). In AMIL, we adapt the latter family of methods to the MIL setting by only applying them to the positive bags (*i.e.* in the positive bag the attribute can either be present or not, in the negative we know for sure that the attribute is absent).

**Attribute-based ZSL.** Zero-shot learning (ZSL) aims at classifying images into classes that are not seen during training (Lampert et al., 2009). Attributes often act as class embeddings that capture visual similarities between classes for transferring knowledge learned from the seen classes to unseen classes. Many approaches map the visual feature into attribute space, then learn a compatibility function between the ground truth and predicted attributes (Romera-Paredes et al., 2015; Changpinyo et al., 2016; Akata et al., 2015; Zhang et al., 2017; Xu et al., 2022). However, these methods are optimized for classification and tend to perform poorly on attribute prediction (Akata et al., 2015). Most recent state-of-the-art ZSL methods are based on generative approaches that bridge the gap between seen and unseen classes by synthesizing visual samples of the unseen classes (Xian et al., 2018b; Narayan et al., 2020). These methods do not output attribute scores and thus are unsuitable for attribute prediction. In order to overcome the issues stemming from the use of class-level attribute descriptions, such as the disregard for intra-class attribute variability and attribute co-occurrence, some methods propose to make the ZSL classification more robust (Jayaraman & Grauman, 2014; Zhu et al., 2018) while others propose to directly improve attribute prediction by decorrelating attribute representations (Jayaraman et al., 2014; Chen et al., 2017). In this work, we show that AMIL can further benefit from an auxiliary loss that correlates related attributes and decorrelates those that are unrelated.

Direct Attribute Prediction (DAP) (Lampert et al., 2013) and its variants (Suzuki et al., 2014) distinguish themselves from the rest of ZSL methods for only training the attribute prediction model, while the classifier is a fixed function of the class-attribute matrix. This makes it, in principle, suitable for assessing the quality of the predicted attributes, unlike more recent ZSL methods that do not include attribute prediction within their pipeline. However, (Lampert et al., 2013) shows that DAP performs less well on higher quality image-level trained attributes than on those trained with class-level labels, suggesting that DAP is not a good proxy for attribute prediction quality. To solve this, we propose MIL-DAP, a modified version of DAP that takes the MIL assumption into account, in order to tackle the asymmetric noise induced by using class-level labels. We show that the ZSL results of MIL-DAP are strongly correlated with the quality of the predicted attributes, making it an alternative for evaluating attribute prediction when no image-level attributes are available at test time.

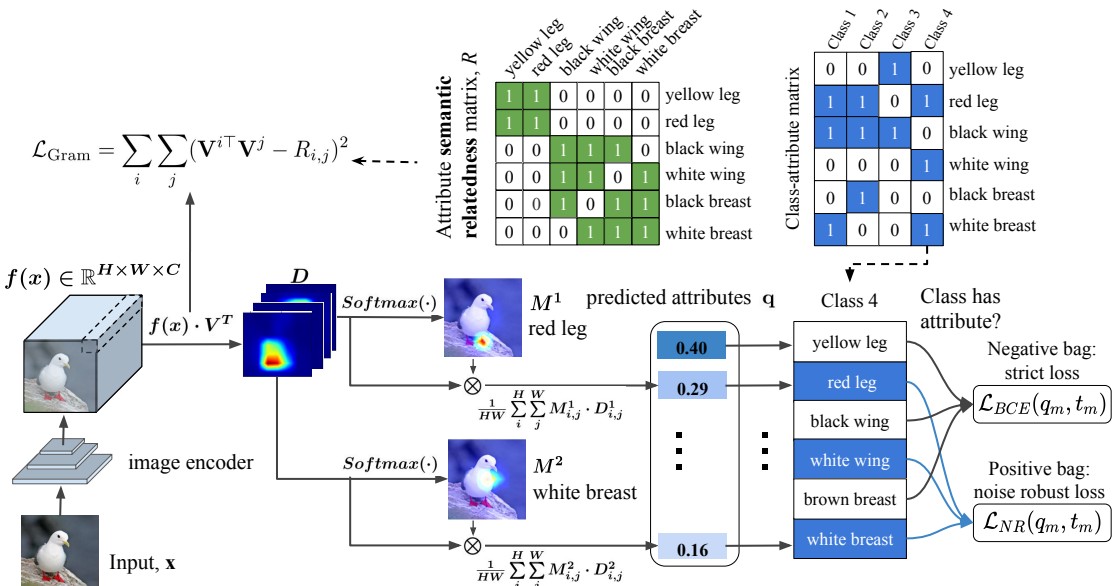

Figure 2: For a given attribute, such as "red legs", we divide the whole dataset in two bags: instances that never display the attribute (negative bag) and those of classes that may (positive bag). A CNN (left) is trained to predict the presence of the attribute. The images from the negative bag are all enforced by a strict binary cross-entropy loss (BCE), but those from the positive bag are learned with a noise robust (NR) loss that is more lenient towards mistakes. In addition, we exploit the information about the semantic relatedness of the attributes by regularizing the attribute prototypes $\mathbf{V}$ such that their Gram matrix closely resembles the matrix of semantic relatedness, $R$.

## 3  Attribute Multiple Instance Learning

Following (Zhao et al., 2019) and as shown in Figure 2 (left), given the input image $\mathbf{x}$ and the CNN encoder $g(\cdot)$, we can get the feature tensor $f(\mathbf{x}) \in \mathbb{R}^{H \times W \times C}$, where $H$, $W$, and $C$ refer to the height, width and channel number. Then the attribute localization module takes image representation $f(\mathbf{x})$ and attribute classifiers $V \in \mathbb{R}^{A \times C}$ as input, and output the dot-product maps $D = f(\mathbf{x}) \cdot V^T \in \mathbb{R}^{H \times W \times A}$, where each map $D^a \in \mathbb{R}^{H \times W}$ encodes the location of the $a^{th}$ attribute. Then we normalize $D$ with spatial softmax, $M = softmax(D)$, to derive the attention map $M^a \in \mathbb{R}^{H \times W}$ for attribute $a$. For instance, the attention map $M^1$ for attribute "red leg" focuses on the leg area, and $M^2$ for attribute "white breast" focuses on the breast.

To predict the presence value for an attribute, we multiply the attention map $M^a$ back to the dot-product maps $D^a$, resulting in an attention weighted feature map, and apply global average pooling as follows:

$$\hat{q}^a = \frac{1}{HW} \sum_{i=1}^{H} \sum_{j=1}^{W} M_{i,j}^a \cdot D_{i,j}^a. \tag{1}$$

We evaluate the attribute prediction results in the ZSL setting, since it ensures that class prediction is not being internally used as a proxy for attribute detection.

### 3.1  The MIL assumption and AMIL

In the weakly-supervised, binary classification-based MIL setting, the dataset is divided in bags, such that one bag $X = (\{\mathbf{x}_1, \ldots, \mathbf{x}_n\}, t)$ contains $n$ instances $\mathbf{x}_j \in \mathcal{X}$. Each instance $\mathbf{x}_j$ has an implicit and unknown

target label $t_j \in \{0, 1\}$, such that the target label of the bag is

$$t = \begin{cases} 0 & \Leftrightarrow t_j = 0 \quad \forall j \in \{1, 2, \ldots, n\}, \\ 1 & \text{otherwise,} \end{cases} \tag{2}$$

meaning that a positive bag must contain at least one positive instance. If the instance labels were known, the bag label could be inferred as $t = \max_{j=1}^{n} t_j$.

In our AMIL setting, we want to learn the presence of specific attributes in images using only class-level labels, along with prior knowledge about relations between attributes. To accommodate this, we extend the MIML (Zhou et al., 2012) framework to *attribute bags* with high overlap between attributes, such that bags can be arbitrarily large. We construct as many pairs of data bags (one positive and one negative) as there are attributes, and each pair of bags will cover the entire dataset. These bags are built using a *binary attribute-class matrix* indicating which attribute is present in which class. Each positive bag $X_a$ is composed by all images of all classes where an attribute can be observed (*e.g.* for $a = red\ leg$, by all images of red-legged kittiwakes *and* arctic terns etc., irrespective of whether the red legs can be seen in a particular image or not, since we do not have this information). Negative bags contain all images of all species where the attribute is known not to be present.

**Attribute MIL (AMIL).** MIL generally builds bags consisting of individual images, where instances are either patches or pixels within the image (Carbonneau et al., 2018). Since the instance-level ground truth $t_j$ is not available and we have only access to the bag-level label, instance-wise predictions need to be pooled into a bag-level prediction. Handcrafted MIL pooling functions, such as the maximum or mean prediction (Wang et al., 2018) have been used, but learnable pooling functions have been shown to perform better (Ilse et al., 2018).

For a bag to accommodate many more instances than a single batch of data, we propose to frame the MIL problem as one of asymmetric noise, as depicted in Figure 2. Instead of having a MIL pooling function that assigns a weight to each instance by directly comparing it to other instances in the bag, as done in previous works, we allow the model to be more flexible when predicting positive attributes while making sure that all negative attributes are predicted as negative. This can be achieved by applying losses designed for noise robustness (NR) to the positive bags while applying a stricter loss, such as binary cross entropy (BCE), to the negative bags. This allows for batch-based optimization without limiting the size of the bags.

Given a CNN $f$, $\mathbf{q} = f(\mathbf{x})$ is the vector of scores of the predicted attributes for input $\mathbf{x}$ belonging to class $y$, such that $q_m$ can be interpreted as $p(a_m|\mathbf{x})$. Given the class-attribute target $\mathbf{t} \in \{0, 1\}^M$ corresponding to class $y$, we propose to use a loss function

$$\mathcal{L}_{\text{AMIL}}(\mathbf{q}, \mathbf{t}) = \sum_{1=m}^{M} \mathcal{L}_{\text{MIL}}(q_m, t_m) \tag{3}$$

such that each element of the loss is

$$\mathcal{L}_{\text{MIL}}(q_m, t_m) = \begin{cases} \mathcal{L}_{\text{NR}}(q_m, t_m), & \text{if } y \text{ has attribute } a_m \\ \mathcal{L}_{\text{BCE}}(q_m, t_m), & \text{otherwise,} \end{cases} \tag{4}$$

where $\mathcal{L}_{\text{BCE}}$ is the binary cross entropy loss and $\mathcal{L}_{\text{NR}}$ is some noise robust loss.

### 3.2 MIL-DAP Model

**Direct Attribute Prediction (Lampert et al., 2013) (DAP).** Given an image $\mathbf{x}$ and an unseen class $y$, we want to estimate $p(y|\mathbf{x})$ by making use of an intermediate attribute representation $\mathbf{a} \in \mathcal{A} = \{0, 1\}^M$ in a two step manner by computing $p(y|\mathbf{x}) = p(y|\mathbf{a})p(\mathbf{a}|\mathbf{x})$. By applying Bayes' rule we obtain:

$$p(y|\mathbf{a}) = \frac{p(\mathbf{a}|y)p(y)}{p(\mathbf{a})}. \tag{5}$$

Note that $p(\mathbf{a}|\mathbf{x})$ can respectively be factorized as:

$$p(\mathbf{a}|\mathbf{x}) = \prod_{m=1}^{M} p(a_m = a_m^y|\mathbf{x}), \tag{6}$$

where

$$p(a_m = a_m^y|\mathbf{x}) = \begin{cases} p(a_m|\mathbf{x}), & \text{if } y \text{ has attribute } a_m \\ 1 - p(a_m|\mathbf{x}), & \text{otherwise,} \end{cases} \tag{7}$$

and that the same can be done for $p(\mathbf{a}|y)$ and $p(\mathbf{a})$. By combining these elements into the first expression we obtain:

$$p(y|\mathbf{x}) = p(y) \prod_{m=1}^{M} \frac{p(a_m = a_m^y|y)p(a_m = a_m^y|\mathbf{x})}{p(a_m = a_m^y)}. \tag{8}$$

Since the class-attribute matrix is binary, we have that $p(a_m = a_m^y|y) = 1$ for all $m$. If we also assume uniform class priors $p(y)$, we obtain the DAP MAP prediction in (Lampert et al., 2013):

$$f_{\text{DAP}}(\mathbf{x}) = \arg \max_{l=1,\ldots,L} \prod_{m=1}^{M} \frac{p(a_m = a_m^{y_l}|\mathbf{x})}{p(a_m = a_m^y)}. \tag{9}$$

We propose a modified version of DAP that is consistent with the MIL assumption and that provides a better proxy for evaluating attribute prediction through ZSL, thus removing the need for image-level attributes for the evaluation.

**DAP under the MIL assumption (MIL-DAP).** We propose three modifications to DAP to make it suitable for estimating attribute quality. First, we observe that under the MIL assumption only the negative values in the class-attribute matrix are considered to be trustworthy, since the positives are interpreted as allowed and not as required. If $\tilde{\mathcal{M}}^y$ is the set of negative attributes associated to class $y$, the MIL-DAP class probability estimation using Eq. (8) becomes:

$$p(y|\mathbf{x}) = p(y) \prod_{m \in \tilde{\mathcal{M}}^y} \frac{[1 - p(a_m|y)][1 - p(a_m|\mathbf{x})]}{[1 - p(a_m)]}. \tag{10}$$

In addition, we propose to modify the way of estimating both $[1 - p(a_m)]$ and $[1 - p(a_m|y)]$. In DAP $[1 - p(a_m)]$ is computed using the class-attribute matrix. However, we observed substantially improved ZSL results, particularly with attributes learned with image-level labels, by computing $[1 - p(a_m)]$ over the training images. In DAP $p(a_m = a_m^y|y) = 1$ for all the attributes. Even if we are assuming the negative attribute labels to be cleaner than the positive ones, we do not expect them to be perfectly clean. In our experiments, we allow for up to $\sim 10\%$ of noise within the negatives due to the threshold used in creating the class-attribute matrix. If we are testing whether $\mathbf{x}$ belongs to class $y$, we can start by assuming that it is indeed the case and then use the observed score as prior:

$$[1 - p(a_m y|y)] = [1 - p(a_m|\mathbf{x})]. \tag{11}$$

If it turns out that indeed $\mathbf{x}$ belongs to class $y$, or to any class in which attribute $m$ is not allowed, then we would be using a more informed class dependent prior. If the assumption does not hold and $\mathbf{x}$ belongs to a class in which $m$ is allowed, $1 - p(a_m|\mathbf{x})$ will tend to be smaller and thus the wrong class $y$ will be further penalized. This brings us to the final MAP formulation of MIL-DAP:

$$f_{\text{MIL-DAP}}(\mathbf{x}) = \arg \max_{l=1,\ldots,L} \prod_{m \in \mathcal{M}^{y_l}} \frac{(1 - p(a_m|\mathbf{x}))^2}{1 - p(a_m)}. \tag{12}$$

**Exploiting attribute relatedness.** Another source of information that can be used to solve some of the ambiguities brought by the weakly supervised setting is the semantic relatedness of the attributes. For instance, if two attributes relate to the same body part or if they describe the same property of different parts (e.g. colors), we may want our model to know that the vector prototype representing these attributes, i.e. the learned weights in the classifier corresponding to those attributes, should be similar. We use this information to build the matrix of semantic relatedness between all pairs of attributes, $R \in [0,1]^{A \times A}$. We propose to encourage the Gram matrix of the set of the normalized attribute prototypes to approach $R$:

$$\mathcal{L}_{\text{Gram}} = \sum_i \sum_j (V^{i\top} V^j - R_{i,j})^2, \tag{13}$$

where $V^i$ is the prototype, or linear classifier weights, of the $i^{th}$ attribute. This loss will make sure that semantically related attributes will have similar prototype vectors while semantically unrelated ones will tend to become orthogonal.

### 3.3 Attribute quality evaluation

To assess the performance of AMIL on attribute prediction, we evaluate it on image-level attribute prediction, attribute-based zero-shot learning with MIL-DAP and weakly supervised part localization.

**Zero-shot learning.** Zero-shot learning aims at recognizing previously unseen classes by transfering knowledge from seen classes. Attributes are widely used as the class embedding for zero-shot classification. However, existing ZSL methods (Romera-Paredes et al., 2015; Xian et al., 2018a; Akata et al., 2015) make use of continuous annotations and thus treat equally the positive and negative class-level attributes for the novel classes. In order to adapt ZSL to the AMIL assumptions, we match the predicted attributes with the ground truth binary attribute of the unseen classes. To ensure that the ZSL results reflect the quality of the attributes, we do not use a compatibility function to train the network as in previous works (Akata et al., 2015; Xu et al., 2020), but simply train the above explained attribute prediction network. At test time, we use the predicted attribute scores and the binary class-level attribute annotation to obtain the class scores with the proposed MIL-DAP, Eq. (12). We also evaluate the attribute quality under a more realistic setting, i.e. generalized ZSL, which aims at recognizing both the seen and unseen classes during inference.

**Part localization.** Our attribute localization module outputs the attention map $M^a \in \mathbb{R}^{H \times W}$ for each attribute $a$, where the high-activated area indicates the location of that attribute. We turn this into a weakly supervised attribute localization task by comparing the predicted attribute location with the ground truth. We evaluate part localization in the ZSL setting where the train and test classes are disjoint.

## 4 Experiments and discussion

We evaluate AMIL using the image-level attribute annotations where available. Then, we evaluate the learned attributes on attribute-based downstream tasks: zero-shot classification and part localization.

**Datasets.** We use three datasets with attribute annotations: CUB-200-2011 (CUB), SUN Attribute Dataset (SUN) and Animals With Attributes (AwA2). The continuous valued class-attribute matrices of all datasets are binarized such that values lower than 0.1 are assigned to the negative bags, making sure that they will generally be cleaner than the positive bags. The value of 0.1 was chosen due to being close to the average values of the CUB and SUN matrices (0.114 for CUB, 0.098 for SUN), which are the values typically used in the literature. We observed no effect on the results on CUB and SUN stemming from this change. In all experiments we use the train-test splits proposed for ZSL in (Xian et al., 2018a) such that the evaluation is always performed on unseen classes. CUB (Wah et al., 2011) is a fine-grained birds dataset, with 11,788 images from 200 classes, and 312 attributes. An attribute consist of two parts: the name of an attribute ("leg color" or "tail texture" etc), and its value ("red" or "striped"). We consider two attributes $i$ and $j$ to be related, $R_{i,j} = 1$, if they share one of the parts, and unrelated, $R_{i,j} = 0$, if they do not. CUB

| Method | AMIL | G | Image-level attribute evaluation | | | | | | | | Class-level attribute evaluation | | | | | |
|---|---|---|---|---|---|---|---|---|---|---|---|---|---|---|---|---|
| | | | CUB | | | | SUN | | | | CUB | | SUN | | AWA | |
| | | | AP | | AUC | | AP | | AUC | | AP | AUC | AP | AUC | AP | AUC |
| | | | s | u | s | u | s | u | s | u | u | u | u | u | u | u |
| Image-level | | | 33.79 | 30.09 | 80.72 | 77.98 | 51.07 | 47.31 | 93.42 | 92.82 | 63.90 | 82.50 | 65.47 | 86.22 | - | - |
| Class-level | - | | 28.07 | 24.84 | 75.49 | 72.66 | 40.49 | 38.69 | 91.46 | 90.50 | 63.78 | 82.33 | 68.30 | 89.23 | 74.04 | 76.66 |
| | - | ✓ | 28.35 | 25.31 | 76.22 | 73.10 | 40.79 | **39.00** | 91.62 | 90.73 | 65.30 | 83.76 | **68.98** | **89.45** | 73.94 | 76.81 |
| Deep-MIML | - | | 28.01 | 24.84 | 75.43 | 72.75 | 40.40 | 38.24 | 91.49 | 90.52 | 63.90 | 82.42 | 68.37 | 89.27 | 74.17 | 76.61 |
| | - | ✓ | 28.39 | 25.17 | 76.44 | 73.03 | 40.72 | 38.95 | 91.59 | 90.71 | 65.31 | 83.77 | 68.86 | 89.29 | 73.94 | 76.81 |
| GCE | | | 27.49 | 23.99 | 74.96 | 72.19 | 39.25 | 37.45 | 91.22 | 90.25 | 63.64 | 81.81 | 68.06 | 89.05 | 74.04 | 76.84 |
| | ✓ | | 28.09 | 24.76 | 74.96 | 72.09 | 39.47 | 38.41 | 91.00 | 90.35 | 63.01 | 81.34 | 67.58 | 88.52 | 73.52 | 75.48 |
| | ✓ | ✓ | 27.92 | 25.14 | 75.30 | 72.94 | 39.72 | 38.83 | 91.27 | 90.50 | 64.17 | 81.89 | 68.25 | 88.53 | 73.74 | 76.34 |
| Bootstrap | | | 27.98 | 24.79 | 75.34 | 72.65 | 39.95 | 38.31 | 91.39 | 90.46 | 63.88 | 82.28 | 68.18 | 89.21 | 73.92 | 76.37 |
| | ✓ | | 28.43 | 25.19 | 76.13 | 72.76 | **41.01** | 38.81 | **91.59** | 90.71 | 64.55 | 83.02 | 68.41 | 89.13 | **74.81** | **76.85** |
| | ✓ | ✓ | **28.68** | **25.62** | **77.33** | **74.43** | 40.59 | 38.92 | 91.59 | **90.87** | **65.97** | **84.33** | 68.68 | 89.29 | 74.79 | 76.78 |

Table 1: Attribute prediction on CUB, SUN and AWA, evaluated with image-level labels and class-level labes. We report average precision (AP) and area under the ROC curve (AUC). AMIL indicates using $\mathcal{L}_{\text{AMIL}}$ and G using the Gram loss. GCE (Zhang & Sabuncu, 2018) and Bootstrap (Reed et al., 2015) represent the two $\mathcal{L}_{\text{NR}}$ we used to test AMIL. Deep-MIML (Feng & Zhou, 2017) is an alternative MIL approach.

is annotated with image-level attributes and body part locations, which we use for evaluation but not for training. SUN (Patterson & Hays, 2012) consists of 14,340 images from 717 scene classes and annotated with 102 scene attributes. AwA2 (Xian et al., 2018a) contains 37,322 images of 50 animal classes with 85 per-class attribute labels, i.e. no per-image attributes are available. Since the attributes of SUN and AwA2 do not follow the binomial structure of CUB, we simply apply a decorrelation loss between all attributes.

**Noise-robust losses and baselines.** In this work we use two methods designed for dealing with noisy labels, i.e. Generalized Cross Entropy (GCE) (Zhang & Sabuncu, 2018) and Bootstrap (Reed et al., 2015). GCE is a generalization of of the Cross-Entropy (CE) and Mean Absolute Error (MAE) losses that aims at finding a trade-off between the noise robustness of MAE and the implicit focus on hard samples of CE, while Bootstrap uses the current model predictions to dilute the ground truth, thus reducing the impact on the loss value of samples for which the model confidently diagrees with the ground truth. We found the truncated version of the GCE loss to work best. With the bootstrapping loss, we obtained the best performance with the soft version (Reed et al., 2015), in which the label is averaged with the current prediction score. In addition we compare with Deep-MIML (Feng & Zhou, 2017) which operates in the traditional, batch-wise, MIL setting. We compare each noise robust loss in the standard setting (i.e. applied to all attributes) and the modified version of Bootstrap and GCE that is adapted to MIL by applying Eq. (4) (GCE+MIL and Bootstrap+MIL). As a baseline, we use the standard approach of training directly on the class-level attributes, while the image-level attributes are used to train a model that serves as an upper bound. In order to test the effect of the Gram loss (Eq. (13)) on the different methods, we apply it both on the proposed appraoches (GCE+MIL+G and Bootstrap+MIL+G) and in the baselines (Class-level Attr+G and Deep-MIML+G).

**Implementation details.** In all the experiments we use ResNet50 pretrained on ImageNet as the image encoder. On all experiments we trained, validated hyperparameters and tested the model using the ZSL train-val-test splits recommended in (Xian et al., 2018a). The attribute prediction results are thus evaluated only on unseen classes. All models are trained for three epochs with a multi-label binary cross-entropy loss or a noise robust loss using the binary class-level label with the exception of the image-level trained upper-bound models for CUB and SUN, which use only BCE. We use the Adam optimizer with a learning rate of 0.0001 for the attribute prediction base model and 0.001 for learning the last linear layer of the attribute prediction model, with a learning rate decay of 0.25 after each epoch.

### 4.1   Attribute prediction results

We start by comparing the different approaches directly on the task of attribute prediction. We report two metrics for multi-label classification: average precision or area under the precision-recall curve (AP) and area under the ROC curve (AUC), both computed per attribute and macro-averaged. We first show the results of using the image-level attributes for training and use it as an upper bound for the class-level trained models, including baselines, the AMIL approach (+MIL) and the Gram loss (+G).

We first evaluate at test time on the image-level attribute labels for CUB and SUN. All methods are trained on the attributes of the ZSL seen classes provided by (Xian et al., 2018a) and tested on the image-level labels of both seen and unseen classes. We observe from Table 1 (left) that the image-level upper bound does indeed reach a substantially better result than any of the class-level methods. We can also see that the proposed scheme tends to improve the results in all metrics, although a large gap to the image-level upper bound is still present. This improvement is much more consistent in CUB than in SUN, where AMIL performs only marginally better than the class-level baseline (e.g. 38.92 AP versus 38.69 AP in the unseen classes). We suspect this is due to the fact that CUB attributes were created with class discrimination in mind, resulting in more distinct class-level attribute representations that lead to better attribute prediction. This can be seen in the fact that class-level AP results are closer to the image level results in CUB (around 5 AP difference) than in SUN (around 10 AP difference). Although this observation does not apply to AUC, this metric seems to be surprisingly saturated in the case of SUN, potentially obscuring the effect of the class-level attributes quality. The methods for learning with noisy labels (GCE and Bootstrap) perform similarly and close to the class-level and Deep-MIML methods. This may be due to the fact that these methods assume similar levels of noise in the positive and negative bags, when the positive bag are likely noisier. This is supported by the fact that, applying the noise robustness only to the positive bags, according to the proposed AMIL, tends to improve performances across the board, showing that the MIL assumptions are indeed met in class-level attribute prediction on these datasets.

We also test on the binarized class-level labels for all datasets, Table 1 (right). Although AMIL also provides an improvement in CUB, results in SUN and AwA2 are more mixed. Most interestingly, the image-level upper bound performs substantially worse than most of the class-level methods, suggesting that evaluating on class-level attributes does not actually reflect the quality of the predicted attributes.

### 4.2   (Generalized) Zero-shot Learning

We evaluate the attribute prediction models using ZSL and generalized ZSL (GZSL) results as proxy metrics. This can be useful in cases where image-level attribute annotations are not available even at test time. We use MIL-DAP, the proposed hard-coded attribute-based classifier in Eq. (12), and show that it better reflects the quality of the predicted attributes. We would like to emphasize that we use ZSL based on attribute prediction as a proxy for attribute quality and this paper does not aim at improving ZSL performance over state-of-the-art ZSL methods, with most methods mentioned in Section 2 reaching substantially higher ZSL accuracy than what is presented here.

GZSL results are shown in Table 2 (left). The first interesting observation in CUB and SUN is that all class-level methods perform better than the image-level upper bound on the seen classes, while the upper bound is best on the unseen classes. This suggests that the class-level models tend to learn the attribute signature of each class rather than the actual attributes, which indicates we should only use the unseen class performance, or the harmonic mean, to assess the quality of the attributes. Indeed, as for attribute prediction, AMIL obtains an improvement over the baselines on CUB and SUN, with mixed results on AwA2, e.g. MIL+G improves the harmonic mean of GCE and Bootstrap on CUB dataset by 5.43% and 5.14%.

ZSL results (Table 2, right) show a similar trend to GZSL on the unseen classes, with the image-level models for CUB and SUN obtaining the best scores and with AMIL resulting in improved results in most cases, e.g. MIL+G improves the accuracy on CUB (by 6.4%) and SUN (by 2.7%) when coupled with GCE.

| Method | AMIL | G | Generalized Zero-shot learning CUB s | u | h | SUN s | u | h | AWA s | u | h | Zero-shot learning CUB | SUN | AWA |
|---|---|---|---|---|---|---|---|---|---|---|---|---|---|---|
| Image-level | - | - | 37.73 | 22.78 | 28.41 | 10.06 | 8.54 | 9.24 | - | - | - | 49.68 | 36.04 | - |
| Class-level | - |   | **59.02** | 8.70 | 15.16 | 13.34 | 5.63 | 7.91 | 93.25 | **1.00** | **1.99** | 36.77 | 33.61 | 36.82 |
| | - | ✓ | 53.28 | 11.19 | 18.50 | 14.06 | 5.28 | 7.67 | 93.76 | 0.61 | 1.22 | 36.40 | 33.68 | 36.67 |
| Deep-MIML | - |   | 58.46 | 8.43 | 14.73 | 12.78 | 5.83 | 8.01 | 93.28 | 0.69 | 1.36 | 35.79 | **33.89** | 37.63 |
| | - | ✓ | 53.05 | 10.92 | 18.11 | 13.80 | 5.83 | 8.20 | 93.68 | 0.73 | 1.46 | 35.83 | 33.47 | 36.11 |
| GCE |   |   | 49.70 | 9.03 | 15.29 | 10.86 | 5.42 | 7.23 | **93.36** | 0.71 | 1.41 | 30.91 | 30.42 | 37.36 |
| | ✓ |   | 55.29 | 9.61 | 16.37 | 12.74 | 5.63 | 7.80 | 93.04 | 0.69 | 1.36 | 36.33 | 32.43 | 37.43 |
| | ✓ | ✓ | 49.33 | **13.11** | **20.72** | 14.02 | 5.35 | 7.74 | 93.09 | 0.56 | 1.12 | 37.31 | 33.13 | 36.33 |
| Bootstrap |   |   | 58.76 | 8.80 | 15.30 | 14.02 | 6.11 | 8.51 | 93.36 | 0.56 | 1.12 | 34.98 | 33.33 | 37.43 |
| | ✓ |   | 58.02 | 10.31 | 17.51 | 14.10 | 5.69 | 8.11 | 92.29 | 0.51 | 1.02 | 39.60 | **33.89** | 37.68 |
| | ✓ | ✓ | 48.59 | 12.94 | 20.44 | **14.78** | **6.39** | **8.92** | 92.80 | 0.32 | 0.63 | **41.52** | **33.89** | **38.27** |

Table 2: Generalized ZSL and ZSL results using MIL-DAP, on CUB, SUN and AWA. For GZSL we report the accuracies on the seen (s) and unseen classes (u), as well as the harmonic mean (h) of the two. For ZSL, we train the attribute prediction network with class-level binary attributes and test it on the unseen classes. Accuracy reported as percentage OA. AMIL indicates using $\mathcal{L}_{\text{AMIL}}$ and G using the Gram loss. GCE (Zhang & Sabuncu, 2018) and Bootstrap (Reed et al., 2015) represent the two $\mathcal{L}_{\text{NR}}$ we used to test AMIL. Deep-MIML (Feng & Zhou, 2017) is an alternative MIL approach.

| Method | AMIL | G | CUB Part Localization Head | Breast | Belly | Back | Wing | Tail | Leg | Avg |
|---|---|---|---|---|---|---|---|---|---|---|
| Image-level | - | - | 66.1 | 67.6 | 49.3 | 32.1 | 37.7 | 10.5 | 44.5 | 44.0 |
| Class-level | - |   | 48.1 | 61.8 | 55.2 | 32.0 | 43.7 | 1.8 | 36.2 | 39.8 |
| | - | ✓ | 56.1 | 62.7 | 62.1 | **35.5** | 59.0 | 2.0 | 45.5 | **46.1** |
| Deep-MIML | - | - | 48.4 | **64.2** | 53.8 | 33.4 | 43.5 | 1.6 | 35.3 | 40.0 |
| | - | ✓ | 56.2 | 62.1 | 62.1 | 34.7 | 57.2 | 2.1 | 46.6 | 45.9 |
| GCE |   |   | 46.8 | 59.1 | 59.1 | 29.4 | 36.2 | 1.5 | 38.2 | 38.6 |
| | ✓ |   | 47.7 | 56.8 | 56.5 | 30.9 | 46.0 | 2.0 | 40.1 | 40.0 |
| | ✓ | ✓ | 51.0 | 59.0 | 62.3 | 33.4 | **60.4** | 2.3 | 49.4 | 45.4 |
| Boot |   |   | 49.3 | 63.7 | 54.2 | 32.6 | 42.8 | 1.7 | 34.9 | 39.9 |
| | ✓ |   | 46.9 | 57.9 | 54.7 | 31.5 | 49.5 | 1.5 | 33.7 | 39.4 |
| | ✓ | ✓ | **56.4** | 56.5 | **63.2** | 32.1 | 55.2 | **2.9** | **50.1** | 45.2 |

Table 3: The Percentage of Correctly Localized Parts (PCP) in CUB. We report the PCP for seven bird body parts as well as the average accuracy. AMIL indicates using $\mathcal{L}_{\text{AMIL}}$ and G using the Gram loss. GCE (Zhang & Sabuncu, 2018) and Bootstrap (Reed et al., 2015) represent the two $\mathcal{L}_{\text{NR}}$ we used to test AMIL. Deep-MIML (Feng & Zhou, 2017) is an alternative MIL approach.

## 4.3 Evaluating Part Localization

A better attribute prediction may also result in an improved learning signal for part localization. Since each attribute in CUB dataset is associated with a body part ("belly", "wing", etc.), we evaluate the part localization ability with the per-image part location annotations. We use directly the attribute attention maps $M^a \in \mathbb{R}^{H \times W}$ (Figure 2, left) for localization. We calculate the Percentage of Correctly Localized Parts (PCP) following the approach of (Xu et al., 2020). The quantitative results on part localization are shown in Table 3, where we observe similar trends as in the previous tasks. However, there are stark differences between the parts. All methods seem to perform well on "Breast" and "Belly", even reaching image-level performance. This is because breast and belly are large parts that usually occupy the center

of the image, making those parts attended to by default, and do not require the model to perform part detection but merely bird detection. The class-level qualitative results in Fig. 3 illustrate the bias towards the center of the bird. We observe that peripheral parts tend to obtain much worse results for the state of the art methods, e.g. the failure at locating "tail", with the highest PCP of 2.9% using Bootstrap+MIL+G. Compared to the other settings, Bootstrap+MIL+G is able to provide usable results on "leg", obtaining an impressive 50.1% PCP, 14 points higher than the baseline. We hypothesize that this increase is likely associated to the positive relations in the Gram loss due to the synergy of learning multiple "leg" prototypes jointly that are able to guide each other towards the right direction.

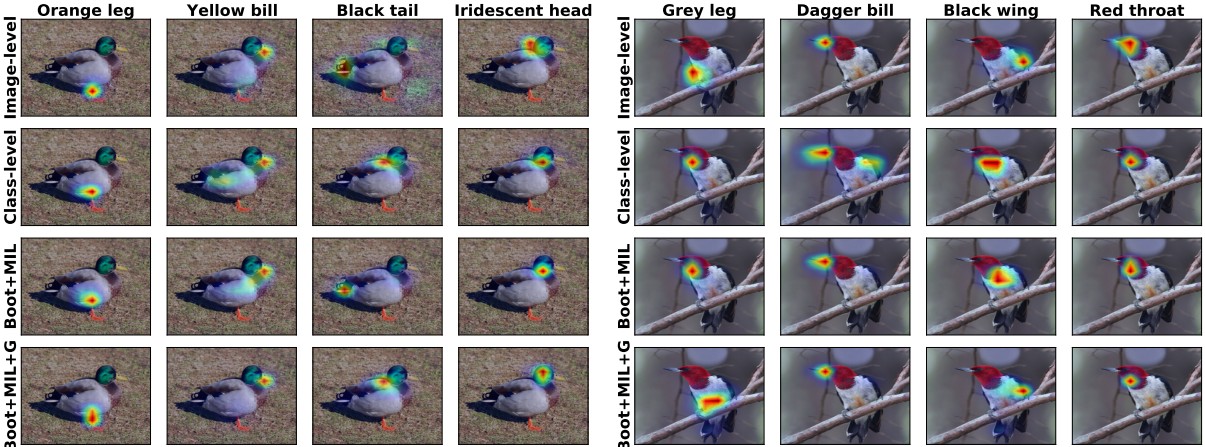

Figure 3: Qualitative part localization on a CUB image displaying the attention maps learned by each model. The $1^{st}$ row corresponds to the image-level upper bound results, the $2^{nd}$ to the class-level baseline, the $3^{rd}$ and $4^{th}$ to the proposed AMIL, with and w/o the Gram loss.

### 4.4 Ablation Studies

**Ablation results on the AMIL loss.** The main assumption for our work, namely the relatively higher reliability of the negative class-level attributes, is captured by the definition of the asymmetric $L_{\text{AMIL}}$ loss in Eq. (4). If this assumption holds, we would expect the results of applying only $L_{\text{NR}}$ or only $L_{\text{BCE}}$ to be worse than using $L_{\text{AMIL}}$. The results of applying only $L_{\text{NR}}$, instantiated as either GCE or Bootstrap (Tables 1), and $L_{\text{BCE}}$, the class-level baseline, show that indeed $L_{\text{AMIL}}$ always results in better image-level attributes prediction than using either $L_{\text{NR}}$ or $L_{\text{BCE}}$. This behaviour is clearer when using the Bootstrap loss, and results are more mixed with the GCE loss. Interestingly, we see comparable results for both ablated settings (using only $L_{\text{NR}}$ or $L_{\text{BCE}}$) and a consistent improvement by applying them separately to the positive and negative samples according to $L_{\text{AMIL}}$ (*AMIL* column in the tables). A similar trend can be observed in the unseen GZSL results in Table 2. This confirms that the MIL assumption of clean negatives and noisy positives does hold in the studied cases.

**Ablation study on the interaction between AMIL and MIL-DAP components.** We explore the contributions of each component of the AMIL and MIL-DAP on CUB and SUN. In the first row of Table 4 we can see how each component of MIL-DAP (i.e. the empirical $p(a)$ prior, the MIL assumption by only accounting for negative attributes and the empirical $p(a|y)$ prior) results in an increase in generalized ZSL performance using the attributes trained with image-level labels. The following three rows show that this trend is absent in the models trained with class-level labels, even when using a noise robust loss (i.e. Bootstrap (Reed et al., 2015)) and the Gram loss, although the latter does result in substantially improved performances in all setting for CUB.

The last two rows show that AMIL, both with and without the Gram loss, results in generalized ZSL performance trends that closely match those of the image-level upper bound. This suggests that the improved

results in attribute prediction obtained by AMIL do indeed result in learning attributes that more closely follow the structure of the image-level attributes. Once again, the application of the Gram loss results in improvements on all settings.

| | **CUB** DAP | +$p(a)$ | +MIL | +$p(a\|y)$ | **SUN** DAP | +$p(a)$ | +MIL | +$p(a\|y)$ |
|---|---|---|---|---|---|---|---|---|
| Image-level | 3.01 | 19.75 | 22.35 | 28.41 | 2.18 | 6.45 | 6.66 | 9.24 |
| Class-level | 12.94 | 14.19 | 15.85 | 15.16 | 8.82 | 8.90 | 8.73 | 7.91 |
| Boot. (Reed et al., 2015) | 13.33 | 14.15 | 16.32 | 15.30 | 9.48 | 9.55 | 9.38 | 8.51 |
| Boot.+G | 20.90 | 22.37 | 22.68 | 20.44 | 8.77 | 9.04 | 8.53 | 7.45 |
| Boot.+MIL | 11.26 | 14.37 | 16.92 | 17.51 | 7.21 | 8.11 | 8.21 | 8.11 |
| Boot.+MIL+G | 12.81 | 17.67 | 18.41 | 20.44 | 6.56 | 8.41 | 8.30 | 8.92 |

Table 4: Generalized zero-shot learning results, harmonic mean between seen and unseen classes, on CUB and SUN. Each row presents an additional modification to DAP (Lampert et al., 2013), with the last row corresponding to MIL-DAP: substituting the attribute prior by an empirical estimate over the training samples (+$p(a)$), applying DAP only using the negative classes (+MIL) and using the attribute prediction of the test image as a prior for the attribute given the class (+$p(a|y)$).

### 4.5 Limitations

Our results show that AMIL and the Gram loss make a substantial difference in the quality of the attributes learned on CUB, where the attributes are all visual and their relatedness is well defined, and that MIL-DAP is useful for detecting this improvement in the absence of image-level labels. In SUN, where some attributes are subjective, the improvement is less marked and in AwA2, where many of the attributes are not visual, the proposed approach seems to result only in a marginal improvement in ZSL and no improvement in GZSL. This suggests that AMIL and MIL-DAP are only appropriate when dealing with clearly visual attributes and that no improvement should be expected for attributes that are subjective or not visual in nature.

## 5 Conclusion

Despite the recent interest in attribute-based methods for applications such as zero-shot classification and explainability, we have observed a large gap in performance between attribute detection trained with weak class-level annotations, as is often done in the literature due to its lower cost of acquisition, and the potential performance that could be obtained by using labour intensive image-level annotations. To close that gap, we propose to leverage the Multiple Instance Learning assumption. We show that applying it to existing noise robustness approaches results in better attribute detection results since it induces tolerance of positive attributes that are not present or occluded on a particular image. Via extensive experiments on three datasets, we show that the proposed approach helps to improve the performance on attribute prediction, and that this improvement can be also be measured via attribute-based tasks such as zero-shot classification and weakly supervised part localization.

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
