# OpenReview forum: "Attribute Prediction as Multiple Instance Learning"
_TMLR — Accepted by TMLR_

### Review · Reviewer_Q5u8 · 2022-05-27

**Summary Of Contributions:**

This paper proposes the Attribute Multiple Instance Learning task based on Multiple Instance Learning task, and utilizes the attention mechanism and positive-negative loss to accomplish attribute prediction and localization. In addition, the superiority of constructive attributes is further explored by improving the DAP model in the ZSL task.

**Broader Impact Concerns:**

I have no more concerns.

**Requested Changes:**

1.	The authors need to add clarification to what is missing in Disadvantages.
2.	The authors need to supplement some experiments, the ZSL models are no longer limited to some outdated methods such as DAP, etc.
3.	Revision of the title to better fit the content of this paper.
4.	The authors should strengthen the connection between ZSL and attribute prediction tasks, which are now somewhat fragmented in current manuscript.


**Strengths And Weaknesses:**

Strengths :
1.	For the improved part of the DAP model, the paper has a more complete theoretical derivation and introduction.
2.	The paper presents the complete ablation studies and the proof of excellence for certain methods. Especially after adding their own innovative part, the effect can get some improvement.
3.	This paper is fully structured and is able to explain their ideas effectively.
Weaknesses:
1.	Some of the content is not presented comprehensively enough. For example, how is the attribute semantic relatedness matrix R constructed; what are the detailed formula definitions of BCE Loss and NR Loss? The authors just directly cite the relevant papers, but do not introduce them.
2.	In the section on positive-negative bags, the authors only explain how to construct them, but do not reflect how they are handled during the training phase.
3.	This paper performs attributes prediction before performing the ZSL task, so there is a discrepancy between the title and the main body of this paper.
4.	The ResNet-101 is recommended as the backbone, as it is the most popular backbone in the ZSL and GZSL tasks.
5.	Why the authors do not utilize generative ZSL methods to test the generated class-level attributes, because the DAP and some remaining linear optimization methods are obsolete in the ZSL field and they are difficult to clearly demonstrate the superiority of the authors' methods.
6.	There are some typos in this paper, for example, in the last paragraph of Sect. Introduction, the authors misspell ‘ours can’ as ‘ours can can’.

---

> ### Author Response · Authors · 2022-06-20
> **Response to reviewer Q5u8**
>
> We thank the reviewer for the time and effort dedicated to this manuscript. In the following we respond the weak points raised by the reviewer.
>
> 1, 2 ) We agree that some additional details on the noise robust losses and training details could render the paper more comprehensive. We will update the manuscript accordingly.
>
> 3, 5 ) We also agree with the reviewer that the main objective of the paper is to perform attribute prediction. We will therefore change the title to:
> “Attribute prediction as Multiple Instance Learning”.
> Our main contribution is the observation that applying the MIL assumption to methods aimed at learning with noisy labels results in better attribute prediction when only class-level labels are available, which is a common setting. Our second contribution is a ZSL method that is able to better measure the quality of the predicted attributes. We thus do not aim at improving ZSL performance, but to align it with attribute prediction. Indeed, we show that the proposed MIL-DAP model is a much better proxy for attribute quality than the original DAP. Most of the generative models directly use the provided class-level attributes as input to synthesize image features, and train a Softmax classifier afterwards [1a] [2a]. We are not aware of any other ZSL methods that make direct use of the attribute predictions and that could therefore be suitable for this task. We will rewrite parts of the manuscript in order to more clearly communicate this connection between  attribute learning and ZSL in the paper.
>
> [1a] Yongqin, Xian, et al. “Feature generating networks for zero-shot learning.” CVPR. 2018.
>
> [2a] Narayan, Sanath, et al. “Latent embedding feedback and discriminative features for zero-shot classification.” ECCV. 2020.
>
> 4 ) With respect to the backbone, we performed some preliminary tests using ResNet models of different depth and ResNet50 seemed to provide a good tradeoff between performance and required resources. For instance, the model trained with image-level attributes results in an attribute prediction AP on the unseen classes of 30.09 with ResNet50 and 30.27 with ResNet101, the class-level baseline 24.84 and 24.89, Deep-MIML 24.84 and 24.92, and Bootstrap+MIL+G results in 25.62 and 25.97 with ResNet50 and ResNet101 respectively. We observe the exact same trends with both backbones, leading to the same conclusions, and only marginally better results with ResNet101, while ResNet50 provides higher efficiency in terms of both time and energy consumption. We would like to point towards the fact that all experiments performed for this paper used the same backbone.

---

### Review · Reviewer_1ko8 · 2022-06-05

**Summary Of Contributions:**

This paper introduces multiple instance learning into attribute learning (AMIL). Specifically, given the asymmetric character of positive attributes and negative ones, the authors propose to under-predict the positive attributes and not to over-predict the negative ones. Based on the AMIL, the authors also proposed an attribute-based zero-shot classification method and verify its effectiveness on three standard benchmark datasets.


**Broader Impact Concerns:**

None.

**Requested Changes:**

See weaknesses.

**Strengths And Weaknesses:**

Strengths:
1. It is interesting to apply MIL to weakly-supervised attribute learning.
2. For positive and negative attributes, the authors adopted two different loss functions, which solves the problem of limited bag size.
3. The authors proposed the MIL-DAP model to apply the proposed method to zero-shot learning.
4. Experiments on benchmark datasets justify its effectiveness.

Weaknesses:
1. It seems that most of the compared methods are proposed several years ago (around 2017). Therefore, I am wondering whether there are any new competitors proposed recently?

2. Missing ablation:  the positive and negative losses in Equ (4) should be ablated individually. For example, what is the result if we only adopt the loss for positive attributes?

3. It seems that the model trained with the image level attributes is the performance upbound. However, in Table 1 and Table 2, the model trained with weakly supervised supervision usually outperforms such upbound. Could explain it?

4. There are some typos:
    P2: '...memory footprint, ours can can'  --> '...memory footprint, ours can'
    Equ (4): it is the definition of L_{MIL} instead of the L_{AMIL}, right?
    In the caption of Table 1, the authors stated that the overall accuracy (OA) is reported. However, in Table 1, the OA results are missing.
    P9: "binarized class-level labels for for all datasets" --> "binarized class-level labels for all datasets"

---

> ### Author Response · Authors · 2022-06-20
> **Response to reviewer 1ko8**
>
> We would like to start by thanking the reviewer for the time taken to review the paper and for the positive comments. In the following we respond to the four raised points.
>
> 1) We thank the reviewer for the comment. After thorough literature research, we were not able to find additional methods that would be suitable in the context of our approach. With respect to ZSL methods, there are many more recent methods, but not related to attribute prediction. To the best of our knowledge, DAP is the most appropriate comparison for our proposition.
>
> 2) We would like to note that using only the negative loss corresponds to the “Class-level Attr” baseline in the tables, while the positive loss corresponds to each noise robust loss without the “+MIL”.
>
> 3) It is expected that training with image-level attributes would result in the best possible results when evaluating on the image-level attributes. The fact that this upper bound model does not outperform all of the class-level models when evaluating with class-level attributes shows that this type of evaluation does not reflect the quality of the attribute prediction. Either image-level attributes or the proposed MIL-DAP have to be used in order to adequately assess attribute quality.
>
> 4) We thank the reviewer for pointing out the typos.

---

### Review · Reviewer_Htzo · 2022-06-13

**Summary Of Contributions:**

The paper proposes a novel MIL-DAP method for attribute prediction in the context of zero-shot learning, which shows its effectiveness on multiple popular benchmarks.

**Requested Changes:**

The following items need to be addressed:
1. clarify the main contributions of this paper, the authors are suggested to provide one paragraph at the end of the introduction
2. more recent methods should be included to make the proposed methods more convincing
3. more details of the noise robust loss should be provided in Sec. 3.1 and ablation studies on selection of such a loss are recommended

**Strengths And Weaknesses:**

The strengths are as follows:
+ the idea of asymmetric class-level attributes sounds reasonable
+ comprehensive evaluation on multiple datasets of different vision tasks
+ easy to follow

The weaknesses are as follows:
- the key contributions of this paper are not sufficiently novel; if I understand correctly,  only using independent losses on positive and negative bags in the context of MIL is the main contribution, while the remaining parts are adopted from existing methods
- comparative methods are a bit outdated, more recent methods after 2018 are suggested to compare

---

> ### Author Response · Authors · 2022-06-20
> **Response to reviewer Htzo**
>
> We thank the reviewer for the time and effort invested in this review and for the positive comments. In the following we address the requested changes.
>
> 1) The reviewer is correct in stating that our key contribution is the use of two losses simultaneously in the context of MIL. However, we would like to highlight that we show the usefulness of such an approach for attribute learning with class-level ground truth, a task that, in our opinion, has not yet attracted the interest it deserves given its potential for making computer vision models more versatile and interpretable. We would like to note that we also propose a new method for evaluating the quality of attribute prediction based on ZSL, thus bypassing the need for image-level attributes even for evaluation purposes. We will clearly summarize our contributions at the end of the introduction in order to clarify this. In addition, we would like to point towards reviewer Q5u8's comments on novelty:
> "For the improved part of the DAP model, the paper has a more complete theoretical derivation and introduction. 2. The paper presents the complete ablation studies and the proof of excellence for certain methods. Especially after adding their own innovative part, the effect can get some improvement."
>
> 2) After thorough literature research, we were not able to find additional methods that would be suitable in the context of our approach. With respect to ZSL methods, there are many more recent methods, but not related to attribute prediction. Most of the more recent ZSL methods are based on generative models that directly use the provided class-level attributes as input to synthesize image features, and train a Softmax classifier afterwards [1a] [2a]. We are not aware of any other ZSL methods that make direct use of the attribute predictions and that could therefore be suitable for this task. To the best of our knowledge, DAP is the most appropriate comparison for our proposition.
>
> [1a] Yongqin, Xian, et al. “Feature generating networks for zero-shot learning.” CVPR. 2018.
>
> [2a] Narayan, Sanath, et al. “Latent embedding feedback and discriminative features for zero-shot classification.” ECCV. 2020.
>
> 3) We will include details about each of the noise robust losses used in the paper, since we agree with the reviewer that this would make the paper more self-contained. Regarding ablation studies, we would like to note that using only the negative loss corresponds to the “Class-level Attr” baseline in the tables, while the positive loss corresponds to each noise robust loss without the methods with the “+MIL” suffix.

---

### Review · Reviewer_M5yj · 2022-06-13

**Summary Of Contributions:**

This paper proposed to modify the attribute learning loss by taking into consideration that positive attribute may not necessarily appear in all training samples from the seen classes. A gram loss is further introduced to exploit the attribute semantic relatedness by encouraging classifier weights of highly related attributes to be close.

**Broader Impact Concerns:**

No concerns.

**Requested Changes:**

1. In particular, I would like to see the justification of gram loss and comparison with alternative designs. This is critical to the decision.

2. It is also good to provide analysis of why performance is not improved much on SUN and AWA datasets.

3. Please provide more analysis into the noise robust loss in the methodology section.

4. Typos: "ours can can handle bags of unlimited" -> "ours can handle bags of unlimited". Please carefully check the caption of Table 4. I find it does not align with the contents in the table. For example, "Each row presents an additional modification to DAP". Is this correct? $p(a|y)$ appears in the table, but it is referred to as $p(a|z)$ in the caption.

**Strengths And Weaknesses:**

Strength:

1. Exploiting the positive bag labels and differentiating positive and negative attributes for zero-shot learning is interesting.

2. ZSL performance is improved by combining with the proposed loss function.

Weakness:

1. The proposed gram matrix loss does not seem to improve under all circumstances. For the method of Deep-MIML I am wondering why there is no comparison of with and without gram matrix loss?

2. Performance improvement is not very obvious for SUN and AWA. Can the author provide some analysis?

3. The motivation of introducing the gram loss is not clear. The semantic relatedness does not guarantee being visually similar. For example, yellow leg and red leg have high similarity, but encouraging the classifier for yellow leg and red leg to be the same does not make sense. Moreover, the gram loss will encourage the features of yellow leg and red leg to be close which essentially harms the generalization capability of the network to unseen classes which may have attributes violating the semantic relatedness calculated from seen classes only.

4. From a technical perspective, alternative gram matrix loss design should be studied as well. Since $\mathbf{V}$ is the classifier weights, the product $f(x)\mathbf{V}^\top$ returns the unbounded logit. Therefore, the scale of $\mathbf{V}$ is unbounded and encouraging the inner product of $\mathbf{V}$ to be 0 or 1 is not reasonable. A more reasonable design could be using the sigmoid cross-entropy loss, i.e. $L_{Gram}=-\sum_i\sum_j R_{ij}\log \mathbf{V}_i^\top\mathbf{V}_j$. Please explain why the regression loss is optimal or provide empirical evaluation by comparing alternative designs.

5. Noise robust loss is the key element in the new formulation. However, it is only briefly mentioned in the experiment section. I am wondering what specific design makes this design better suited for positive bags of attributes.

---

> ### Author Response · Authors · 2022-06-21
> **Response to reviewer M5yj**
>
> We sincerely thank the reviewer for thoroughly reading our paper and for the provided comments. In the following we respond to the points raised in the weaknesses section.
>
> 1) We would like to note that we report the results of all methods, including Deep-MIML, both with and without the gram loss. In all tables, +G indicates with the Gram loss, and we will emphasize this in the revised manuscript. The Gram loss is generally improving the performance of attribute prediction, ZSL and attribute localization. In Table 2, though adding Grad loss decreases the seen class accuracy (s), it significantly improves the unseen accuracy (u) and results in higher harmonic mean (h). For Deep-MIML we see that the Gram loss provides a small but consistent boost in performance, although generally smaller than with the proposed approach.
>
> 2) CUB has been designed with attribute characterisation in mind, so we expect it to be the best showcase method for an approach like ours. In SUN and AWA, the relations between attributes and classes is less explicit and sought for, but we still observe improvements, showing that caring about attributes is important to improve image classification tasks as well. We can see that the class-level attribute representation in SUN is further apart from the image-level representation when we compare it to CUB. This is shown by a much larger drop in image-level performance when training with class-level labels in SUN than in CUB. We will include this discussion in the revised manuscript.
>
> 3) We would like to note that the semantic relatedness of attributes is independent of the classes. "Red leg" and "yellow leg" are considered to be related because they relate to the same part (leg). Our hypothesis is that both "red leg" and "yellow leg" detectors need to be sensitive to the visual features present in legs of any color. Indeed the reviewer is right in saying that we cannot always assume that semantic relatedness means visual similarity, but this may often be the case for visual attributes. All legs in birds share some visual commonalities (in terms of shape and relative position in the body) and all red parts share some clear visual similarities.
>
> 4) With respect to alternative ways of implementing the Gram loss, we indeed found them substantially less performing than the regression approach. We would like to note that V is not unbounded, since each vector is normalized to unit norm before computing the inner product similarities. Although this is mentioned before Eq. (13) in the manuscript, we will emphasize this in the revised manuscript. A Gram loss based on sigmoid cross-entropy is something we did consider. We assume the reviewer is suggesting to use a binary cross entropy loss with sigmoid activations $$\mathbf{L}_{Gram} = -\sum_i\sum_j R^{ij} \log(\sigma(\mathbf{V}_i^\top \mathbf{V}_j)) - \sum_i\sum_j(1-R^{ij})\log(1-\sigma(\mathbf{V}_i^ \top \mathbf{V}_j)),$$ where $\sigma(\cdot)$ is the sigmoid function, since using only the first part of the expression only encourages related attributes to become similar, without encouraging unrelated ones to become dissimilar. However, in our experiments, it resulted in a degradation of the performance, with the Bootstrap+MIL+G method obtaining 24.81 AP on the unseen classes in CUB, down from 25.62 when using the regression-based Gram loss. We think this is down to the fact that, due to the use of a sigmoid activation, negative similarities are encouraged for unrelated attributes, leading to negatively correlated attribute representations rather than the desired decorrelation.
>
> 5) We agree that the paper would benefit from a detailed description of each noise robust loss and we will include them in the revised manuscript.
>
> We also thank the reviewer for pointing out the typos, including the mentioned mistake in the notation.

---

### Decision · Action_Editors · 2022-07-28

**Recommendation:** Accept with minor revision

**Comment:**

The paper studies attribute learning under the setting of multiple instance learning, where only the image-level labels are available. The paper presents a few technical contributions and validates them in the problem of zero-shot learning. Initially some common weakness
points are raised by more than one reviewers, including the lack of comparisons with more recent ZSL methods, and the less comprehensive
ablation studies. The authors have successfully addressed most of these raised concerns. Two of three reviewers who have given final ratings are mostly convinced by the responses. Given the reviews and discussions, acceptance with mandatory revision is decided. The authors are expected to revise and improve the paper by addressing all the comments from the reviewers and including them in the final paper. More precisely, the authors are expected to improve the paper by (1) presenting comparisons with more recent ZSL methods, even though many of these methods may not involve attribute prediction, (2) including more ablation studies as requested by reviewers (e.g., those on positive and negative losses), (3) change of the paper title appropriately, (4) other typo issues pointed out by the reviewers. With the revised paper, the authors are required to include a "Summary of Change" file that explains clearly how the paper has been revised according to the above comments.